Influence of tied-ridge-furrow with inorganic fertilizer on grain yield across semiarid regions of Asia and Africa: A meta-analysis

http://orcid.org/0000-0003-3832-1259 Mak-Mensah Erastus 1
Obour Peter Bilson 2
http://orcid.org/0000-0002-5517-5807 Wang Qi 1 wangqigsau@gmail.com
1 College of Grassland Science, Gansu Agricultural University , Lanzhou , China
2 Department of Geography and Resource Development, University of Ghana , Accra , Ghana
Okpala Charles
Electronic publication date: 2021 Aug 17
Publication date: 2021
Volume: 9
Electronic Location ID: e11904
Received 2021 Apr 8; Accepted 2021 Jul 13
Copyright: © 2021 Mak-Mensah et al.
Copyright year: 2021
Copyright holder: Mak-Mensah et al.
License: This is an open access article distributed under the terms of the Creative Commons Attribution License, which permits unrestricted use, distribution, reproduction and adaptation in any medium and for any purpose provided that it is properly attributed. For attribution, the original author(s), title, publication source (PeerJ) and either DOI or URL of the article must be cited.
License URL: https://creativecommons.org/licenses/by/4.0/

Keywords: Rainwater harvesting, Tied-ridging, Mulching, Fertilizer, Grain yield

Funding: National Natural Science Foundation of China 42061050 and 41661059 This research was funded by the National Natural Science Foundation of China (42061050 and 41661059). The funders had no role in study design, data collection and analysis, decision to publish, or preparation of the manuscript.

==============================
Background

In semiarid areas, low productivity of crops has been attributed to lack of appropriate soil moisture conservation practices since droughts and soil erosion are rampant in most areas of this region. Consequently, ridge-furrow rainwater harvesting is widely used in these regions across the globe. Despite ridge-furrow being widely practiced, tied-ridge-furrow has not been extensively adopted by small-scale farmers in semi-arid regions. Consequently, the effectiveness of tied-ridge-furrow as a viable method of increasing crop yield has received less attention.

Methodology

For large-scale implementation, a detailed assessment of how ridge furrow, tied–ridge-furrow with fertilizer, tied-ridge-furrow with mulching and tied-ridge-furrow without mulching or fertilizer influence crop yield in different agro-environments under varying climatic conditions is needed. This study used the PRISMA guidelines to determine the impact of tied-ridge-furrow rainwater harvesting technique with mulching or fertilizer on sorghum (Sorghum bicolor) and pearl millet (Pennisetum glaucum) grain yields.

Results

Sorghum grain yield increased by 17% greater in tied-ridge-furrow without mulching or fertilizer in comparison to flat planting. This may be due to increase in soil organic carbon in the region (9 g kg−1). Grain yield of millet significantly increased by 20–40% in Africa from 18 study observations in tied-ridge-furrow with fertilizer application as compared to tied-ridge-furrow without mulching or fertilizer treatments. This might be due to the significant increase in total nitrogen by 13–42% in the soil at <50 mg kg−1 quantity which had an effect size of 469.14 [65.60, 872.67]. In terms of soil texture, grain yield of millet and sorghum significantly increased in heavy textured soils (clay loam, silt clay, and clay soils) with an effect size of 469.14 [65.60, 872.67] compared to light and medium-textured soils of zero effect sizes. Millet and sorghum grain yields in tied-ridge-furrow with mulching, on the other hand, were not significantly different from those in flat planting. This may be due to the mulching materials used in those tests.

Conclusion

In view of yields of sorghum and millet increased significantly by 32% and 17% in tied-ridge-furrow without mulching or fertilizer treatment compared to flat planting and tied-ridge-furrow with fertilizer treatment compared with tied-ridge-furrow without mulching or fertilizer treatment, respectively, this study recommend the use of fertilizers in a tied-ridge-furrow system to increase grain yield in semiarid areas compared to flat planting. Again, the study recommends more research on tied-ridge-furrow systems with other organic mulches and fertilizers in semiarid areas.

Introduction

Globally, droughts characterized by low soil water availability and poor soil fertility are major factors limiting plant growth and yield (Olsovska et al., 2016). The problem is more severe in semiarid areas, which had been home to grain legumes (Hashem et al., 2019). This is due to the steepness of slopes and erosive nature of rainfall which make soil erosion severe in semi-arid lands (Wolka, Mulder & Biazin, 2018). Erosion depletes soil organic matter, which in turn decreases aggregate soil stability and water holding capacity which increases soil crusting (Wolka, Mulder & Biazin, 2018). In view of providing a lasting solution to this challenge, Mo et al. (2016) proposed an innovative, low-input, and high-yield farming practice called ridge-furrow rainwater harvesting system (RFRWHs).

In rain-fed agriculture, the RFRWH method has been beneficial and widely adopted by local farmers to increase crop water use efficiency (Eldoma et al., 2016; Zhang et al., 2018; Pan et al., 2019). The furrows created in RFRWH system can be left open, or closed at regular intervals for holding water and facilitating infiltration. Tied-ridges are formed in the field when the furrows are blocked with earth ties at predetermined distances to create a series of micro-catchment basins (Biazin & Stroosnijder, 2012). Tied-ridging offers maximum potential for water conservation which can be useful to plants for a longer period (Ndlangamandla, Ndlela & Manyatsi, 2016). Although RFRWH is a widespread tillage practice, tied-ridging has not been widely adopted by small-scale farmers in semi-arid regions (Jensen et al., 2003). Meanwhile, the adoption of soil moisture conservation techniques such as tied-ridges and mulching is reported to improve soil moisture retention (Ndlangamandla, Ndlela & Manyatsi, 2016). Mulching could save 25% of water used in maize and wheat cultivation in rain-fed agriculture, according to Yan et al. (2015). Thus, there is an urgent need to explore improved water use and soil fertility management practices for optimal crop productivity in rain-fed agriculture in China (Andrew Tapiwa, 2019).

Besides drought, another factor constraining productivity in rain-fed agriculture is low soil fertility (Deng et al., 2006). The main ways to improve soil fertility include increasing soil organic matter by increasing the percentage of legume and green manure crops and combining this with the use of inorganic fertilizers. The use of crop residues as mulch and integrating them into the soil after crop harvest increases soil organic matter and fertilizer use efficiency (Deng et al., 2006). Inorganic fertilizers when applied as plant food, increases soil pH in the short-term, but can increase soil acidity in the long term due to nitrification of ammonium (Bährle-Rapp, 2007). Therefore, understanding how plant biomass worldwide, particularly in Asia and Africa, change with fertilizers and mulching applications in tied ridges is vital to harnessing their potential for large-scale implementation in agricultural production (Antala et al., 2020). Consequently, a more accurate method is required to estimate the effect sizes of fertilizer and mulch application practices in tied ridge-furrow on crop yield, WUE, and soil physical properties. Hence, meta-analysis was conducted to shed light on the influences of these practices, elucidates their impacts, and provides a new and more vigorous theoretical model.

Meta-analysis, fundamentally the ‘analysis of analyses’, is a method of quantitatively (a) recognizing overviews from a range of separate and disparate studies, and (b) determining inadequacies in existing research such that new preferences for future research can be proposed (Kuznetsov, Passot & Sulem, 2008). Meta-analysis allows for statistical analysis of effect sizes and objective evaluation of other authors’ experimental results (Mak-Mensah et al., 2021). Meta-analysis increases the statistical potential for evaluating theories and comparing treatment differences in various contexts (Luo, Wang & Sun, 2010). The effect size observed in each sample can be assumed to be an unbiased estimate of the underlying true effect size, subject to random variance. Meta-analysis can clarify trends in a quantitative way that in conventional reviews might be perceived as being biased by personal judgement (Verheijen et al., 2010). Research outcomes are coded into essential classifications for comprehensive discussion of both the status of scientific understanding on a specific ‘effect’, possible underlying mechanisms and marginal or exceptional conditions (Kuznetsov, Passot & Sulem, 2008). Since so much depends on the quality of results that are to be synthesized, there is the danger that adherents may simply multiply the inadequacies of the data base and the limits of the sample (e.g., trying to compare the incomparable) (Kuznetsov, Passot & Sulem, 2008). As new studies are published, meta-analyses on effect of tied-ridge-furrow with inorganic fertilizer on grain yield across semiarid regions of Asia and Africa can be updated (and refined) periodically once a large enough body of research has been established (Verheijen et al., 2010).

Although some field experiments have been conducted to assess the impact of fertilizer, yield and mulching use relationship in ridge-furrow rainwater harvesting system, no comprehensive, and quantitative analyses of available published data have been conducted thus far. Therefore, investigating grain yield across semiarid regions of Asia and Africa under the influence of tied-ridge-furrow with inorganic fertilizer is one of great importance to sustaining rain-fed agriculture on the Loess Plateau. Through a synthesis of existing results, the current study examined the impacts of tied-ridge-furrow with inorganic fertilizer on grain yield. The objectives of this study were (1) to quantify the responses of grain yield to fertilizer and mulching and establish the yield-fertilizer or mulching use relationship on the Loess Plateau and (2) to compare the effects of different tied-ridge-furrow systems with or without fertilizer and mulching methods on yield. To achieve these objectives, the PRISMA guideline (Moher et al., 2009) was used to perform a meta-analysis on related literature to determine the effect of inorganic fertilizer in tied-ridge-furrow on grain yield and soil physical properties.

Materials & methods

Data collection

Scope of peer-reviewed papers

Peer-reviewed papers published in English between 2000 and 2020 that investigated the effects of tied-ridge-furrow with or without mulching or fertilizer on field crops were retrieved from online databases (ISI Web of Science, PubMed, Google Scholar, Scopus (Elsevier), JSTOR, and Science Direct) as previously described in Mak-Mensah et al. (2021). More than one database was used to minimize selection bias.

Literature search strategy

The literature search for the meta-analysis (MA) was done using the following keywords: ‘rainwater harvesting’, and/or ‘ridge furrow’, and/or ‘tied ridge’, and/or ‘mulching’ and/or ‘yield’. The search yielded 101 publications, which were screened based on the following inclusion criteria: (1) on-field experimentation with at least ridge-furrow, tied-ridge-furrow, mulched ridges, fertilizer, flat planting, and no mulch treatments; (2) experimental fields recorded were located in rain-fed semiarid agricultural areas, and (3) crop yield was reported.

Inclusion and exclusion criteria

Based on the inclusion criteria, 85 of the papers found were excluded from this meta-analysis. As such, 16 papers were subjected to the study (Table 1). The publication screening procedure was adapted from the PRISMA meta-analysis protocol and is illustrated in a flowchart (Fig. 1) (Moher et al., 2009).

Figure 1 Flowchart of literature identification, and screening for use in this study. Adapted from PRISMA (Moher et al., 2009).

Table 1 Study areas, crops and literature sources used in this meta-analysis.

References	Cardinal point (N, E, m a.s.l)	Crops	Location	Country	NOS	
Berhanu, Beshir & Lakew (2020)	12°68′, 39°15′, 1,976	Pearl millet	Sekota	Ethiopia	8	
Aleminew et al. (2020)	39°63′, 12°15′, 1,512	Kobo	Ethiopia	7	
Sibhatu et al. (2017)	12°41′50″, 39°42′08″, 1,578	Sorghum	Fachagama	Ethiopia	9	
Zelelew, Ayimute & Melesse (2018)	39°04′, 12°63′, 2,254	Sekota	Ethiopia	7	
Brhane et al. (2006)	138°14′06″, 388°58′50″, 1500	Abergelle	Ethiopia	6	
Grum et al. (2017)	13°52′49″, 39°28′59″, 2,408	Maize	Gule	Ethiopia	6	
Ademe, Bekele & Gebremichael (2018)	N/A	Sankurra	Ethiopia	8	
Adeboye et al. (2017)	70°33′0″, 40°34′0″, 271	Soybeans	Ile-Ife	Nigeria	9	
Pan et al. (2019)	41°08′22.8″, 111°17′43.6″, 1589	Sunflower	Wuchuan	China	6	
Dong et al. (2017)	43°33′ and 125°38′/42°30′, 118°88′	Maize	Shuangyang (Jilin/Chifeng)	China	7	
Ren et al. (2017)	35°15′, 110°18′, 850	Ganjing	China	8	
Xiukang, Zhanbin & Yingying (2015)	35°12′, 107°40′, 1,206	Changwu	China	9	
Liu & Siddique (2015)	36°02′, 104°25′, 2,400	Zhonglianchuan	China	6	
Song et al. (2013)	43°30′23, 124°48′34, 220	Gongzhuling	China	6	
Wang et al. (2011)	113°39′, 34°43′, 111.3	Zhengzhou	China	7	
Wang et al. (2014)	35°29′, 107°45′, 1,264	Ningxian	China	7	
Note:

NOS stands for Newcastle Ottawa Scale.

Heterogeneity occurrences and uncertainties

Heterogeneities may have occurred from the following factors: (1) the researchers had different preferences and personal experiences; (2) many of the field experiments did not include any long-term observations; (3) the experimental fields exhibited different yield statuses, mulching and fertilizer application rates before sowing; and (4) field management practices and climatic conditions differed during crop-growing seasons. Although the effect of tied-ridge-furrow with fertilizer and mulching can be evaluated through meta-analytical method, underlying sources of meta-analytical uncertainties require further research. Figures 2–4 depict locations of field experiments for studies included in the MA. The data from selected papers were classified based on biophysical parameters determined (Table 2). Table 3 shows mean, range, and coefficient of variation (CV) of yield for millet (Pennisetum glaucum) and sorghum (Sorghum bicolor) in different ridge-furrow rainwater harvesting practices with or without mulching and fertilizer, while Table 4 shows locations and precipitations.

Figure 2 Experimental locations from the peer-reviewed publications in Ethiopia for the meta-analysis. ArcGIS 10.6 software (ESRI, Redlands, California) was used to produce the map.

Figure 3 Experimental locations from the peer-reviewed publications in Nigeria for the meta-analysis. ArcGIS 10.6 software (ESRI, Redlands, California) was used to produce the map.

Figure 4 Experimental locations from the peer-reviewed publications in China for the meta-analysis. ArcGIS 10.6 software (ESRI, Redlands, California) was used to produce the map.

Table 2 Categorization of data within the selected publications.

Annual mean precipitation	Annual air temperature	Organic C content	Soil bulk density (0-20 cm)	Soil texture (0–20 cm)	pH	Soil available N	Soil available P	FC	Tillage management	
≤500 mm	<10 °C	<9 g/kg	<1.3
g cm−3	Light: sandy and sandy loam soils	Very acidic:
pH < 5	<50
mg kg−1	<20
mg kg−1	<150
mg kg−1	RF, ridge–furrow;	
>500 mm	10–20 °C	>9 g/kg	>1.3
g cm−3	Medium: loamy sand and loam soils	Acidic:
pH 5–6	>50
mg kg−1	>20
mg kg−1	>150
mg kg−1	TRFS, tied ridge–furrow with fertilizer;	
	>20 °C			Heavy: clay loam, silty clay, and clay soils	Neutral:
pH 6–7				TRFP, tied ridge–furrow with mulching	
					Slightly alkaline:
>7					
Notes:

<500 (low mean precipitation); >500 mm (high mean precipitation); <10 °C (low mean temperature); 10–20 °C; >20 °C (high mean temperature); <9 g/kg (low organic C content); >9 g/kg (high organic C content); <1.3 (low soil bulk density) g cm−3; >1.3 g cm−3 (high soil bulk density); <5 (acidic pH); Neutral: pH; >6–7, acidic: pH 5–6, very, >7–8 (slightly alkaline); <50 (low soil available N) mg kg−1; >50 mg kg−1 (high soil available N); <20 (low soil available P) mg kg−1; >20 mg kg−1(high soil available P); ≤25% (low FC); >25% (high FC).

Table 3 Mean, range, and coefficient of variance (CV) of grain yield in the various tillage systems with fertilizer and mulching combinations.

Treatment	Mean	Total	Range	CV	
Tied ridge with mulching	2610.0	2	–	–	
	2155.0	2	2110–2200	2.95	
Tied ridge with fertilizer	3157.2	2	2851–3463	9.69	
	3896.1	5	2537.6–5653	30.81	
	2790.0	2	–	–	
	3846.2	2	3153.4–4539	25.47	
Tied ridge	2622.0	2	1831–3413	42.66	
	2743.2	5	2648–2839.8	2.67	
	3394.0	2	2653.6–4134.5	30.85	
	1914.2	2	1680–2540	28.82	
	2685.0	2	2500–2870	9.74	
	8350.0	4	7700–9000	6.52	
	1415.0	3	860.3–2301.8	54.84	
Ridge-furrow with mulching	12277.5	4	8026–14021	23.37	
	1456.7	3	0–2742	94.67	
	9480.5	4	8217–10601	11.65	
	3364.9	3	3330.7–3407.2	1.16	
	8564.0	2	3726–13402	79.89	
	6863.8	2	6142–7585.6	14.87	
	7043.3	3	5440–8600	22.44	
Ridge-furrow	11247.3	4	6581-13140	27.76	
	1079.7	3	0–2005	93.67	
	6850.5	2	414–13287	132.87	
	10670.0	3	8503.3–11949.6	17.68	
	5933.3	2	4941–6925.5	23.65	
Flat planting	9758.8	4	5014–11660	32.70	
	1036.3	3	0–2231	108.45	
	8246.0	4	7255–8844	8.87	
	3711.4	3	3627.1–3795.7	2.27	
	6747.0	2	378–13116	133.50	
	10419.8	3	8543.6–11614.7	15.79	
	5503.2	2	4805–6201.4	17.94	
	6600.0	3	6190–6940	5.76	
	1988.0	4	1061–3339	51.80	
	2954.4	10	1789.6–4086	27.26	
	2805.0	2	–	–	
	2901.4	4	2107.7–3870.4	28.65	
	1600.0	2	1560–1640	3.54	
	1135.0	2	790–1480	42.99	
	6150.0	4	5400–6600	8.55	
	750.5	3	0–1749.1	119.99	

Table 4 Heterogeneity analysis on grain yield under flat planting compared to tied ridge, tied ridge compared to tied ridge with fertilizer, and flat planting compared to tied ridge with mulching using random-effects models.

			Heterogeneity		
Tillage combinations	Categories	n	Chi2	df	P	I2%	Z	Overall effect (P)	
Flat planting compared
to tied ridge	Africa	20	34	6	<0.00001	82	1.98	0.05	
Sorghum	7	2.1	2	0.35	5	3.63	0.0003	
Millet	7	0.75	1	0.39	0	0.62	0.54	
Temperature (>20 °C)	6	0.12	2	0.94	0	1.15	0.25	
Precipitation (>500 mm)	6	0.12	2	0.94	0	1.15	0.25	
Soil texture (heavy)	14	14.5	4	0.01	72	1.34	0.18	
Soil organic matter (<10)	7	0.75	1	0.39	0	0.62	0.54	
pH (>6–7)	9	14.2	2	0	86	0.9	0.37	
Soil organic carbon (<9 g kg−1)	4	0.8	1	0.37	0	3.91	<0.0001	
Total Nitrogen (<50 mg kg−1)	14	14.5	4	0.01	72	1.34	0.18	
Phosphorus (<20 mg kg−1)	9	14.2	2	0	86	0.9	0.37	
Field capacity (>25%)	5	0.02	1	0.87	0	1.12	0.26	
Permanent wilting point (>10)	5	0.02	1	0.87	0	1.12	0.26	
Tied ridge compared
to tied ridge with fertilizer	Africa	9	0.6	2	0.74	0	2.28	0.02	
Millet	7	0.6	1	0.44	0	2.23	0.03	
Temperature (>20 °C)	4	0	1	0.97	0	1.91	0.06	
Precipitation (>500 mm)	4	0	1	0.97	0	1.91	0.06	
Soil texture (Heavy)	9	0.6	2	0.74	0	2.28	0.02	
Soil organic matter (<10)	7	0.6	1	0.44	0	2.23	0.03	
pH (>6–7)	7	0.17	1	0.68	0	1.47	0.14	
Total Nitrogen (< 50 mg kg−1)	9	0.6	2	0.74	0	2.28	0.02	
Total Phosphorus (< 20 mg kg−1)	7	0.17	1	0.68	0	1.47	0.14	
Flat planting compared
to tied ridge with mulching	Africa	4	4.4	1	0.04	77	0.71	0.48	
Temperature (>20 °C)	4	4.4	1	0.04	77	0.71	0.48	
Soil texture (light)	4	4.4	1	0.04	77	0.71	0.48	

Statistical analysis and meta-analysis

The importance of publications examined in this analysis was determined using the Newcastle Ottawa Scale (NOS) (Zeng et al., 2015). A 7-point scale was used to assess high-quality publications (papers). The NOS scores ranged from 6 to 9 on a scale of one to ten (Table 1). Since more precise measurements have a greater effect on the overall sample (Yu et al., 2018), results from research with more accurate measurements are given more weight.

We used construction confidence interval analysis to correlate the severity of the response ratio between the ridge-furrow, tied-ridge-furrow, and flat planting treatments (Gao et al., 2019). According to Gao et al. (2019) and Qin, Hu & Oenema (2015), we measured the effect size as the normal log (ln R) of the response ratio (R) which reflects the severity of the effect of ridge-furrow and tied-ridge-furrow on yield in this meta-analysis (Hedges, Gurevitch & Curtis, 1999), Eq. (1):

(1) R=θt/θc,

(2) In=In(θt/θc)=Inθt−Inθc,

where θt and θc are mean values of yield in tied-ridge-furrow and flat planting, respectively. According to Li et al. (2018), yield percentage change (Z) was calculated to further validate results of this study:

(3) Z=(R−1)×100%

where a negative percentage change indicates a decrease in tied-ridge-furrow variable when compared to flat planting, and a positive percentage change indicates an increase in the corresponding variable when compared to flat planting. Subsequently, in addition to the means, sample sizes of the variables and standard deviation (SD) were extracted from the papers or estimated using the equation (Yu et al., 2018):

(4) SD=SE×√n

For studies that did not record SD, average coefficient of variation (CV) within each data set was calculated, and the unavailable SD was approximated using the following equation (Yu et al., 2018):

(5) SD=CVA~−θ

where θ is average of flat or tied-ridge-furrow planting with mulch or fertilizer. Since effect sizes of tied-ridge-furrow with mulching or fertilizer and flat planting for crop yield are continuous variables, random-effects models were implemented using the Nordic Cochrane Centre’s Review Manager Program (RevMan; ver. 5.3, Denmark). The heterogeneity between studies was assessed in this analysis using Chi2 and I2 statistics (Table 4). The I2 test value of: I2 < 25% indicates no heterogeneity, 25–75% indicates mild heterogeneity, and I2 > 75% indicates high heterogeneity (Table 4) (Higgins, 2003). In cases with mild to high heterogeneity (Chi2 p-value < 0.05 and X2 > 50%), a random-effects model was used. The ridge-furrow, tied-ridge-furrow with fertilizer, tied-ridge-furrow with mulching, and flat planting groups’ mean differences were weighted according to their SE and sample sizes, and their confidence intervals (CI) were calculated from their weighted effect sizes. When there was no zero in the 95% CIs of the effect size of treatment, the impact of that treatment was significant. Concurrently, when the 95% CIs contain zero, the treatment was not significant. To demonstrate distribution symmetries of individual experiments, frequency distribution of effect sizes (Odds ratio) was calculated using an Excel 2010 spreadsheet.

Results

General characterization of findings

Literature search yielded 101 publications (papers), which were screened based on inclusion and exclusion criteria and 16 papers were subjected to the study. China was the only Asian country examined in this study. African countries considered in this study were Ethiopia and Nigeria. This MA focused on the influence of tied-ridge-furrow as a viable method for increasing crop yield. The studies considered in this MA examined sorghum and millet production in Asia and Africa under temperatures > 20 °C, precipitation > 500 mm, soil organic matter < 10, pH > 6 to 7, soil organic carbon < 9 g kg–1, total nitrogen < 50 mg kg−1, phosphorus < 20 mg kg−1, field capacity > 25%, and permanent wilting point > 10. Furthermore, soil textures considered in this MA were sandy and sandy loam soils (light), loamy sand and loam soils (medium), and clay loam, silty clay and clay soils (heavy). Data from most studies analyzed were in tables thus were transferred into the database of this MA directly.

Influence of tied-ridge-furrow system with organic mulching on crop yield

Through a synthesis of existing results, the current study examined impacts of tied-ridge-furrow with inorganic fertilizer on grain yield. Literature search yielded 101 publications (papers), which were screened based on inclusion and exclusion criteria and 16 papers were subjected to the study. China was the only Asian country examined in this study. African countries considered in this study were Ethiopia and Nigeria. In Ethiopia, sorghum is produced in almost all areas occupying an estimated cumulative land area of 1.68 million ha with national average yield of 2,369 kg ha−11. However, most soils are low in fertility mainly due to nutrient mining as about K (27.3 kg ha−1), P (5.9 kg ha−1), and N (22.5 kg ha−1) per annum are lost from pearl millet production. In Nigeria, rainfall unpredictability and recent variations in weather conditions threatens crop yield and farmers income. Since this condition is not different from Ethiopia, a drought tolerant and high yielding sorghum variety, PAN 8625 is widely grown, with an average yield of 5,533.83 kg ha−1. This MA showed that tied-ridge-furrow with mulching is mainly practiced in Africa (Ethiopia and Nigeria) where temperatures are mostly greater than 20 °C (Table 4). Results from this studies revealed there was no significant difference between grain yields in tied-ridge-furrow with mulching compared to flat planting (p = 0.48). Since mulches have a major impact on soils they are applied to, the absence of a significant difference is most likely attributable to the mulching materials used in those tests. The mulching materials used in these studies were guinea grass (Panicum maximum) which was applied on the ridges at 21,600 kg ha−1 and 0.2 m thick and covered with 75% of soil after crop propagation and grass mulch applied at 10 cm thickness.

Yield response of sorghum in tied-ridge-furrow without mulching or fertilizer

When compared to flat planting, increase in sorghum yield in tied-ridge-furrow without mulching or fertilizer fields was significant. In 14 study observations, tied-ridge-furrow with no fertilizer or mulching applied treatment yielded 17% sorghum grains greater than flat planting. Sorghum yield had an effect size of 1,197.92 [551.25, 1,844.60] with a p-value of 0.05 (Fig. 5). This may be because soil organic carbon (9 g kg−1) has increased significantly (p < 0.0001) in this region. The I2 test for sorghum yield in this MA yielded a value of 5%, which was less than 25%, suggesting that the studies used in this analysis were not heterogeneous. The random-effects model was therefore used because Chi2 p-value was < 0.05. However, concerning precipitation, temperature, pH, total nitrogen, phosphorus, soil texture, and soil organic matter, there was no significant difference (p > 0.05) in crop yield in these regions.

Figure 5 Odds ratios of crop yields in (A) flat planting compared to tied ridge. (B) tied ridge compared to tied ridge with fertilizer. The error bars signify 95% confidence intervals, and the values above the bars indicate the number of observations (n).

Yield response of millet in tied-ridge-furrow system with inorganic fertilizer applications

The yield of millet increased 20–40% in Africa (Ethiopia and Nigeria) from 18 study observations in tied-ridge-furrow with fertilizer application as compared to tied-ridge-furrow without mulching or fertilizer treatments (Fig. 5). The effect size of the overall study effect (p = 0.03) was 469.87 [57.75, 881.99]. The I2 for this comparison is 0%, indicating that there is no heterogeneity among the studies included in this study. This might be due to the significant increase in total nitrogen (13–42%) in the soil at < 50 mg kg−1 quantity which had an effect size of 469.14 [65.60, 872.67]. With an effect size of 469.14 [65.60, 872.67], yields in soils with heavy texture (clay loam, silt clay, and clay soils) also increased significantly. This subgroup also had 0% in I2 test, which may have affected the substantial increase in soil organic matter in the area. In this MA, effect size of soil organic matter was 469.87 [57.75, 881.99]. However, concerning precipitation, temperature, pH, total nitrogen, and phosphorus, there was no significant difference in crop yield in these regions.

Discussion

The current study examined impacts of tied-ridge-furrow with inorganic fertilizer and mulching on grain yield through a synthesis of 16 existing publication results. In this MA, crop yield had no significant difference between tied-ridge-furrow with mulching and flat planting (Refer to Table 4), which is consistent with Ndlangamandla, Ndlela & Manyatsi (2016), who found no major differences in sorghum yield between tied-ridge-furrow with mulching and flat planting. According to Demoz (2016), this negative effect is probably due to high soil temperatures created within the ridge which can be detrimental to seed germination, and shallow infiltration of moisture into the soil compared to that on flat soil when rainfall is light. Variations in seasonal rainfall can significantly affect crop yields (Silungwe et al., 2019). Tied ridges produce rectangular pools between ridges, increasing surface retention capacity and reducing runoff, resulting in improved soil moisture content and, as a result, crop development and yields. In Zimbabwe, Motsi, Chuma & Mukamuri (2004) discovered that tied-ridge-furrow increased soil moisture compared to regular tillage, particularly during dry months. Guzha (2004) in a similar study discovered that higher soil moisture was held in ridges and that; this was connected with higher roughness due to ridge shape. In addition, Silungwe et al. (2019) discovered that tied ridges hold soil moisture and increase yields more than flat planting in rain-fed agriculture. Consequently, Belachew & Abera (2014) found that planting maize in tied ridge furrows increased maize yield by 32.3% as compared to flat planting.

However, according to Yoseph (2014a), even though Cowpea (Vigna Unguiculata L.) yield and yield components were not significantly affected by moisture conservation practices in their experimental study, yield advantage of tied-ridge-furrow was 26% compared to flat planting. The grain yield of pearl millet (Pennisetum glaucum) obtained from tied-ridge-furrow (3,634 kg ha−1) was higher by 12.52% compared to farmers’ practice (3,179 kg ha−1) (Yoseph, 2014b). In contrast to flat planting, this MA showed a substantial increase in sorghum yield in tied-ridge-furrow without mulching or fertilizer plots (Refer to Fig. 5). This is consistent with findings of Adeboye et al. (2017), who found that tied-ridge-furrow rainwater harvesting practices increased grain yield by 14.0–41.8% relative to flat planting. According to McHugh et al. (2007), tied-ridge-furrow and no-till significantly reduced seasonal soil loss by up to 11,000 kg ha−1 during seasons with moderate intensity storms. Similar research by Netsere, Kufa & Tesfaye (2015) discovered that compared to flat planting and untied ridge, tied-ridge-furrow increased Arabica Coffee yield by 19.0–23.6%. Local farmers who practiced tied-ridge-furrow realized a statistically significant (p < 0.05) difference in yields of about 3,000 kg ha−1 compared to flat planting whose yields were about 1,500 kg ha−1 (Motsi, Chuma & Mukamuri, 2004). In another study by Zelelew, Ayimute & Melesse (2018), tied–ridge-furrow provided the highest grain yield of 2,300 kg ha−1 compared to flat planting which gave a yield of 1,750 kg ha−1 in the dry season. Again, Milkias, Tadesse & Zeleke (2018) investigated in-situ rainwater harvesting field experiments over two years and reported a significant increase of 143.14% in maize grain yield due to increased soil moisture storage by tied-ridge-furrow compared to flat planting treatment. Tied ridges have higher moisture content at 0–5 cm and 6–10 cm depths than flat planting (Mandumbu et al., 2020).

Conversely, Gerbu (2015) reported that field experiments had a yield advantage of 56–68% in improved varieties with fertilizer and tied-ridge-furrow treatments compared to flat planting of local sorghum cultivar without fertilizer as an attractive option to boost sorghum yield under moisture stress environment. According to this MA, millet grain yield increased by 20% to 40% in tied-ridge-furrow treatments with fertilizer application compared to tied-ridge-furrow without mulching or fertilizer treatments (Refer to Fig. 5). Results here consolidate previous findings of Aleminew et al. (2020), who found highest grain yield (3,355 and 3,145 kg ha−1) for pearl millet, with application of micro-dose fertilizer with dry seed in tied-ridge-furrow and recommended fertilizer rate with dry seed in tied-ridge-furrow, respectively. In similar research, Gebrekidan (2003) found highest yield increment of 1,361 kg ha−1 (34.5%) due to tied-ridge-furrow compared with flat planting on Entisols with NP followed by 1,255 kg ha−1 (48.5%) on Alemaya black clay soils (Vertisols) under fertilized conditions, indicating that yield response to water conservation treatments was higher under fertilized than under unfertilized conditions on the two soils. Accordingly, maximum sorghum grain yield (3,226.70–4,621.00 kg ha−1) under fertilized and (2,678.00–4,318.80 kg ha-1) unfertilized conditions were obtained from closed tied-ridge-furrow with planting in-furrow (Sibhatu et al., 2017). In addition, for maize, they found that highest grain yield (4,414 and 4,392 kg ha-1) recorded was with the application of micro-dose fertilizer with primed seed in tied ridge with intercropping mung bean and recommended fertilizer rate with dry seed in tied-ridge-furrow (Aleminew et al., 2020). This, according to Biazin & Stroosnijder (2012), might be due to tied-ridge-furrow with fertilizer being more effective in improving crop yields during seasons with low rainfall events (280–330 mm). Meanwhile, most farmers in semi-arid areas, due to risk of crop failure and poor harvests emanating from periodic water shortages, investing in fertilizers (and other inputs) is simply not worthwhile (Rockström, Barron & Fox, 2002). This leaves them in low-risk, low-yield and low-income agriculture (Dercon & Christiaensen, 2011). Therefore, alleviating agricultural water deficiency through the use of tied-ridge-furrow may give farmers the conviction to invest in soil enhancement practices (fertilizers or biochar) for improved crop production in rain-fed regions. The authors suggested the use of tied-ridge-furrow with fertilizer application to enhance crop yield thus developing a global agricultural scheme capable of meeting up with food safety and at the same time, achieve economic, social, and environmental sustainability (Okpala, 2020).

Conclusions

Given the poor soil fertility level of semiarid areas in Asia and Africa, a single intervention through rainwater harvesting techniques may not bring a substantial impact on crop productivity. Based on the results of this study, it can be concluded that tied-ridge-furrow rainwater harvesting system with fertilizer application retains soil moisture and could be adopted by farmers in semiarid areas to increase crop yields. While there was no substantial difference in crop yield in tied-ridge-furrow with mulching treatment compared to flat planting, there was a significant increase in sorghum yield in tied-ridge-furrow without mulching or fertilizer fields compared to flat planting. In view of yields of sorghum and millet increased significantly in tied-ridge-furrow without mulching or fertilizer treatment compared to flat planting and tied-ridge-furrow with fertilizer treatment compared with tied-ridge-furrow without mulching or fertilizer treatment, respectively, this study recommend the use of fertilizers with tied-ridge-furrow system to increase yield in semiarid areas compared to flat planting. The study recommends more research on tied-ridge-furrow systems with other organic mulches and fertilizers in semiarid areas.

Supplemental Information

Supplemental Information 1 Raw data.

Click here for additional data file.

Supplemental Information 2 PRISMA checklist.

Click here for additional data file.

Supplemental Information 3 Rationale for conducting the meta-analysis.

Click here for additional data file.

Additional Information and Declarations

Competing Interests

Author Contributions

Data Availability

The authors declare that they have no competing interests.

Erastus Mak-Mensah conceived and designed the experiments, performed the experiments, analyzed the data, prepared figures and/or tables, authored or reviewed drafts of the paper, and approved the final draft.

Peter Bilson Obour performed the experiments, analyzed the data, prepared figures and/or tables, and approved the final draft.

Qi Wang conceived and designed the experiments, authored or reviewed drafts of the paper, and approved the final draft.

The following information was supplied regarding data availability:

The raw measurements are available in the Supplemental File.

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
