# Peer review of "Influence of tied-ridge-furrow with inorganic fertilizer on grain yield across semiarid regions of Asia and Africa: A meta-analysis"

_PeerJ, doi:10.7717/peerj.11904_

## Round 0.1 · original submission · Major Revisions

Reviewers have attended to your work, and have raised some concerns that need your kind attention. Please, address all the comments in detail, point by point response as well. Look forward to your revised manuscript. Thank you very much

Reviewer 1 ·

Basic reporting

The authors of the paper Influence of tied-ridge-furrow with inorganic fertilizer on grain yield across semiarid regions of Asia and Africa: a meta-analysis propose a relevant theme because droughts and soil erosion has been attributed to the problem of appropriate soil moisture conservation practices and lead to lack of crops produced in the mentioned region of this studied.

Experimental design

Given that this is a meta-analysis, before the last paragraph of introduction needs to establish the following:
-What is a meta analysis and how and why is it used?
-Has there been any metaanalysis /systematic review performed in this area of research ? The reason is to clearly justify why this meta analysis is needed, as well as help establish a clear gap that this current work aims to fill
Information written about metaanalysis in the data analysis section Lines 112-117 should be brought here, and further information supplemented, with adequate references
Methodology
The aim of this paper is very interesting. The research design appears well built. However, kindly remove authors names from data collection. Kindly break down the methodology section into the following below, making sure to refer effectively to the flow chart at each stage
- The protocol followed to achieve this meta analysis
-The search strategy must stand out alone, here feel free to engage the challenges encountered in conducting the search itself
-Inclusion and exclusion criterion must be clearly explained
-How was the data extracted, kindly describe this step-wise and succinctly
-Data analysis’ should be changed to ‘statistical analysis and meta analysis’
Results
Given that this is a metanalysis, it is useful to start the results and discussion with:
-Process of conducting the review , this should be the start of line 163 . Kindly
-Characteristics of studies, that has been meta-analysed. These should form the components that have built the meta analysis, it is essential that this has to be characterized
- Meta analysis should then follow, where the influence and yield can be presented
Please clarify and consider if it is possible to provide a short description of parts of Asia and Africa countries that applied the specific approach comparatively. For Instance, West Africa Countries, Line 164 – 165.
These are essential aspects of meta analysis, authors are encouraged to follow this, to improve clarity of this work

Validity of the findings

This work appears to present very valid results. Authors are encouraged to elaborate more on the discussion, and add more information from Tabulated data herein.
The conclusions are properly presented by the authors of the research, reflecting the final results of the comparisons with the existed studies.

Additional comments

Authors have made very strong effort in this work. However, major revisions are required to improve it, towards potential acceptance.

Reviewer 2 ·

Basic reporting

no comment

Experimental design

The hypothesis authors made "using inorganic fertilizers and mulching in semi-arid areas is the best agronomic method for reducing drought's negative effects and avoiding runoff and soil erosion.". I just wandering how authors test a method is best?

Validity of the findings

No novel finding was presented in this MS.

Additional comments

In the MS, authors tried to compare the impact of different cultivated systems for cereal crop yield using a meta-analysis based approach. But the results were not any new discovery and the conclusions were not strongly supported by the results.

In row 82-84: "As such, we hypothesized that using inorganic fertilizers and mulching in semi-arid areas is the best agronomic method for reducing drought's negative effects and avoiding runoff and soil erosion.", authors need to check the "best", how do authors test "inorganic fertilizers and mulching in semi-arid areas is the best agronomic method"?

In the row 165-167, authors presented " temperatures are mostly greater than 20°C (Table 4). Results from the studies revealed there was no significant difference between grain yields in tied ridge with mulching compared to flat planting ", but in row 174-177, authors showed tied-ridge-furrow system significantly increases sorghum yield. The results were different, authors need to explain it.

I don't think the results support the author's conclusion. Such as in row 255-257 "Based on the results of this study, it can be concluded that the combination of tied ridge rainwater harvesting system with mulching treatment was the best technique in retaining soil moisture and can be adopted by farmers in semiarid areas to increase crop yields.", but the results part (row 165-167) showed "Results from the studies revealed there was no significant difference between grain yields in tied ridge with mulching compared to flat planting (p > 0.05)."

In total, there is no difference from previous research, or authors need to figure out what is the new discovery of the MS.

---

## Round 0.2 · Minor Revisions

Authors, kindly find comments raised by the reviewer. In addition, the editor would request attention to these details:

a) Materials and methods should be properly sub-divided, to guide readers, as follows:
- Scope of peer-reviewed papers
-Literature search strategy
- Inclusion and exclusion criteria
-Heterogeneity occurrences and uncertainties
-Statistical analysis and meta-analysis

b) results should start with
-General characterization of findings (here, authors should relay this information {Literature search yielded 101 publications (papers), which were screened based on inclusion and exclusion criteria and 16 papers were subjected to the study. China was the only Asian country examined in this study. African countries considered in this study were Ethiopia, Zimbabwe, Nigeria, and Swaziland} but into more details ok
-Influence of tied-ridge-furrow system with organic mulching on crop yield
-Yield response of sorghum in tied-ridge-furrow without mulching or fertilizer
-Yield response of millet in tied-ridge-furrow system with inorganic fertilizer applications


Please, all references cited in the results should be removed. This is results being relayed, and not results being discussed.

Please, all exact p-values, and corresponding R-sq (adj), and F-values must be reported. Please, all exact values, must be reported, whether it is significant, or not significant. The same applies to heterogeneity as well.

c) Please, in the discussion, where results presented in either Table or Figure is mentioned, please, all must be referred to , as in (Refer to Table ?) (Refer to Figure ?)

Looking forward to your revised manuscript. Thank you.

Reviewer 1 ·

Basic reporting

The authors of the paper Influence of tied-ridge-furrow with inorganic fertilizer on grain yield across semiarid regions of Asia and Africa: a meta-analysis propose a relevant theme because droughts and soil erosion has been attributed to the problem of appropriate soil moisture conservation practices and lead to lack of crops produced in the mentioned region of this studied.

Experimental design

How meta analysis and why it was used, was properly described by the authors before the last paragraph of introduction as requested, appears to clearly justify its relevance for the study. Gaps that also justify the need for the study has been presented.

Validity of the findings

The experiment appear valid.


The methodology has improved. However, it is important that authors further elaborate on the following:
- The methods section has to be broken down into its relevant subsections
An example, sections like inclusion and exclusion criteria, search strategy should all stand out in a subsection.


Results have improved. However, in the discussion, more literature is required to strengthen the discussion. Example, in areas where meta analysis results are being compared.

Additional comments

Look forward to the revised manuscript.

---

## Round 0.3 · Minor Revisions

The authors have made great efforts to revise their work, and address comments raised.

Please, kindly attend to the following:

- This appears to be a typo or mistake, probably from a previous version that has been left in. The results state that "China was the only Asian country examined in this study. African countries considered in this study were Ethiopia, Zimbabwe, Nigeria, and Swaziland." However, table 1 only shows locations for studies from Ethiopia, Nigeria and China. Something is not clear here! Possibly, Zimbabwe and Swaziland might have been left out of the table. Or, should it be removed from the results text? Obviously, there is something missing. Authors, please carefully address these discrepancies.

-Table 1, it might be helpful to remind the readers that NOS stands for Newcastle Ottawa Scale.

- In addition , the nation ‘Swaziland’ is officially renamed to Eswatini as of 2018. It will be useful for authors to consider it, and use it.

Look forward to your revised manuscript. Thank you for your efforts.

---

## Round 0.4 · accepted · Accept

Thank you for addressing all comments raised, and improving your manuscript. The revised manuscript is now acceptable for publication. The authors have benefited from the peer review process, which strengthened the quality of their work. Thank you authors for finding PeerJ as your journal of choice, and look forward to your future scholarly contributions. Congratulations.